# Perineuronal Net Alterations Following Early-Life Stress: Are Microglia Pulling Some Strings?

**DOI:** 10.3390/biom14091087

**Published:** 2024-08-30

**Authors:** Reza Rahimian, Claudia Belliveau, Sophie Simard, Gustavo Turecki, Naguib Mechawar

**Affiliations:** 1McGill Group for Suicide Studies, Douglas Mental Health University Institute, Verdun, QC H4H 1R3, Canada; reza.rahimian@mail.mcgill.ca (R.R.); claudia.belliveau@mail.mcgill.ca (C.B.); sophie.simard2@mail.mcgill.ca (S.S.); gustavo.turecki@mcgill.ca (G.T.); 2Department of Psychiatry, McGill University, Montreal, QC H3A 0G4, Canada; 3Integrated Program in Neuroscience, McGill University, Montreal, QC H3A 0G4, Canada

**Keywords:** microglia, extracellular matrix, perineuronal nets, early-life stress, parvalbumin interneurons

## Abstract

The extracellular matrix plays a key role in synapse formation and in the modulation of synaptic function in the central nervous system. Recent investigations have revealed that microglia, the resident immune cells of the brain, are involved in extracellular matrix remodeling under both physiological and pathological conditions. Moreover, the dysregulation of both innate immune responses and the extracellular matrix has been documented in stress-related psychopathologies as well as in relation to early-life stress. However, the dynamics of microglial regulation of the ECM and how it can be impacted by early-life adversity have been understudied. This brief review provides an overview of the recent literature on this topic, drawing from both animal model and human post mortem studies. Direct and indirect mechanisms through which microglia may regulate the extracellular matrix—including perineuronal nets—are presented and discussed in light of the interactions with other cell types.

## 1. Introduction

Early-life stress (ELS) has enduring effects on brain structure and function and increases the risk of psychiatric disorders later in life [1]. Numerous investigations have shown that stressors during sensitive developmental periods can affect mood, cognition, and neuroimmune pathways. The interactions between stress and inflammation have led the field to focus, using a wide range of approaches, on the microglia in cortico-limbic regions. Moreover, stressors experienced in early life have been found to affect the maturation of local inhibitory circuits and the associated formation of perineuronal nets (PNNs), a form of specialized extracellular matrix (ECM) involved in the closure of critical periods of development [2]. Indeed, changes in cortical PNNs and parvalbumin-positive (PV) interneurons following ELS have been reported in both human post mortem and animal studies [2,3,4]. Animal models have allowed researchers to investigate the interaction between microglia and ECM in physiological and pathological conditions and a few post mortem studies have also explored this interaction in Alzheimer’s and Huntington’s diseases. However, investigations into the relationship between the microglia and the ECM following ELS and in stress-associated psychopathologies are seemingly limited. The objective of this short paper is to review the existing literature on the mechanisms underlying the interactions between the microglia and the ECM while providing a perspective on how ELS can modulate such interactions at the molecular level.

## 2. PNNs in Development and Early-Life Stress

The ECM is a three-dimensional structure composed of collagen, enzymes, and glycoproteins that provides essential support to cells. In the brain, the ECM is primarily composed of glycosaminoglycans (GAGs), unbranched polysaccharide chains made up of repeating chondroitin sulfate-disaccharide units [5]. In 1893, Camilo Golgi described a specific form of ECM in the form of reticular structures enwrapping the cell bodies and largest proximal dendrites of certain neurons in the brain. These structures, known today as PNNs, then became intensely studied until Ramón y Cajal interpreted them as a staining artifact about twenty years later [6]. Renewed curiosity about PNNs emerged in the early 2000s, when they were described as key players in the regulation of the critical period of cortical plasticity during development. These nets, a condensed form of the ECM, are predominantly composed of chondroitin-sulfate proteoglycans (CSPGs) from the lectican family (aggrecan, brevican, neurocan and versican) attached to varying numbers of GAG side chains. While the dynamics of PNN formation are not completely understood, it is recognized that the synthesis and secretion of PNN components occur from neurons themselves as well as from surrounding glial cells [3,7]. The appearance of PNNs encapsulating neurons begins as early as postnatal day (PD) 14 in the rodent cortex [8], and from as early as 2 months after birth in humans [9]. The quantity of PNNs continues to increase until PD 33/40 in rodents [8,10] and until 8–12 years old in humans [9,11], coinciding with the end of the critical period of plasticity. In both rodent and human central nervous systems (CNSs), the highest densities of PNNs are found in the primary motor and somatosensory areas, with lower densities observed in the prefrontal cortex, spinal cord, cerebellum, and subcortical structures, such as the hippocampus and amygdala [12,13]. PNNs predominantly (70–80%) enwrap PV interneurons in the visual, somatosensory, and motor cortices, as well as in the amygdala and hippocampus [3,9,12]. Few studies have been systematic in characterizing the phenotype of the remaining 20–30% of neurons covered by PNNs, most of which seem to be non-PV interneurons. In the human ventromedial prefrontal cortex (vmPFC), in accordance with the rodent literature, 74% of PNNs were found surrounding PV neurons, while an estimated 23% of PNNs were described around non-PV interneurons. In addition, ~3% of PNNs in this region were observed around excitatory neurons, albeit more faintly stained [3]. This is also consistent with previous reports in rodents, which found faint PNNs around pyramidal neurons in the cortex, as well as in the CA2 region of the hippocampus and the amygdala [12].

Loss of function experiments have been key to understanding the role played by PNNs in closing critical periods of cortical plasticity during development and in maintaining limited neuroplasticity in mature cortical circuits. In a pioneering study by the Maffei group, the injection of the bacterial enzyme chondroitinase ABC (chABC—which degrades GAGs) in the visual cortex of adult rodents led to a juvenile-like shift in ocular dominance [14], together with decreased inhibition and an increase in gamma activity [15]. This work provided the insights that the closure of critical periods of plasticity is not a passive phenomenon and that it involves PNNs. The latter also play a crucial role in learning and memory (reviewed in [16]). For example, PNNs limit feedback inhibition from PV cells onto projection neurons in the hippocampus and anterior cingulate cortex (ACC), which in turn enhances the consolidation and reconsolidation of contextual fear memory [17]. Moreover, it has been observed in rodents that the transition in fear memory resilience occurs at the conclusion of the critical period, which coincides with the development and maturation of PNNs. Fear memories formed in adulthood, which are resistant to extinction and associated with post-traumatic stress disorder, are actively protected by PNNs in the amygdala. Degradation of these nets with chABC allows for subsequent fear memories to be extinguished in adult mice [18]. Furthermore, ELS paradigms, such as maternal separation, during the critical period of plasticity can induce a premature shift in fear memory resilience, resulting in juvenile rats exhibiting adult-like resistance to fear memory extinction [19]. This research aligns with the precocious maturation theory of child abuse [20,21], wherein abused children exhibit accelerated puberty.

Although ELS has been reported to have long-lasting effects on PNN development and maturation, the results vary depending on the model organism and ELS paradigm, as well on the brain region examined and the approach used to label PNNs (Table 1) [22]. Umemori and colleagues investigated the effects of perinatal exposure to various substances on brain development, reporting that fluoxetine exposure did not alter the PV cell density or PV expression but resulted in delayed PNN formation in the amygdala and hippocampus, with reduced PNN densities at PD 17 and PD 24 [23]. A 2018 study using a developmental scarcity–adversity mouse model found an enhanced threat response from the basolateral amygdala with a reduction in PV-mediated synaptic inhibition and decreased PNN intensity at weaning [24]. In the same year, Page and colleagues reported that chronic mild unpredictable stress led to a decrease in the PV cell number in the prelimbic cortex of mice, with no effect of ELS on the PNN density [25]. More recently, a study using the repeated maternal separation with early weaning mouse model reported a decreased intensity of PV interneurons along with an increased intensity of PNNs in the ventral hippocampus. Interestingly, no change in density was observed in CA1 of the hippocampus [26]. Using a rat model of maternal separation, Gildawie et al. revealed sex-specific and region-specific effects, notably a decreased density of PNNs in the prelimbic prefrontal cortex of juvenile male mice, which was restored in adulthood [27]. Changes associated with maternal separation in the intensity of PNNs around PV interneurons were noted in the adult prelimbic cortex, with no such effect observed for those surrounding other unspecified cell types. Both males and females exhibited a reduced density of PNNs in the infralimbic cortex in adulthood after maternal separation, with no effect on the intensity of PNN staining. Additionally, there was an increased density of nets in the basolateral amygdala of male adolescent rats but not in juveniles or in adults [27]. These findings, along with increased synaptic plasticity, were replicated by another group in an ELS rat model of limited bedding but were only observed in the right hemisphere of males [28]. More recently, aggrecan-labeled PNNs in the ACC were found to be increased in an early-life social stress mouse model [2]. In view of the variability of the findings reported so far, it becomes all the more important to investigate the impact of ELS in humans. To date, few studies have investigated PNNs in post mortem human brains and, to the best of our knowledge, only three studies from our group, all conducted in adult vmPFC samples, have investigated the potentially lasting impact of child abuse (CA) on PNNs. The first study found a CA-associated increase in PNN density (labeled with Wisteria Floribunda Lectin (WFL)) and somatodendritic coverage of neurons in depressed suicides [3]. The second revealed a three-fold increase in the proportion of unmyelinated PV inhibitory interneurons covered by a PNN and greater PV labeling intensity in myelinated cells covered by PNNs in individuals with a history of CA [29]. Finally, the third study examined the PNN composition measured by liquid chromatography tandem mass-spectrometry combined with PNN labeling by WFL or anti-aggrecan and found no changes in samples from individuals with a history of CA compared to matched controls [30]. In brief, CA seems to lastingly increase the development and enhance the maturation of PNNs in the human vmPFC, with certain subtypes of PV interneurons being particularly affected, but with no accompanying change in the PNN molecular composition. The following section will focus on the microglia as modulators of the ECM, including PNNs, in ELS.

## 3. Microglia in Development and Early-Life Stress

Microglia play multiple roles in normal physiology through interactions with neurons and other glial cells within the developing CNS, notably influencing synaptic maturation and brain wiring [31,32]. In fact, microglia actively contribute to shaping neuronal circuits using different mechanisms [33,34]. The complement system, with C1q and C3 as its most important elements expressed in subsets of immature synapses, is the most studied mechanism. Microglia are the only cell type in the CNS that expresses the C3 receptor (C3R) that can target immature synapses for phagocytosis [35,36]. However, microglia-mediated synaptic pruning is not always complement-dependent. For instance, neuroglial interactions through the CX3C motif chemokine ligand 1 (CX3CL1) and the CX3C chemokine receptor 1 (CX3CR1) play a prominent role in complement-independent synaptic elimination [37,38]. It is noteworthy that most of our knowledge on the importance of microglia during development stems from studies on excitatory neurons [39,40], with only recent work having been conducted on microglial interactions with inhibitory neurons, such as PV interneurons [41]. Considering the fact that PV neurons are ensheathed by PNNs, recent reports of the interaction between microglia and inhibitory neurons shed more light on how ECM-PNNs are modulated by microglia. Favuzzi and collaborators revealed that microglial metabotropic GABA-B receptors selectively shape developing inhibitory circuits in the mouse somatosensory cortex [42], therefore showing that the removal of the GABA-B receptor from microglia changes inhibitory but not excitatory connectivity. The authors also reported that genes related to synaptic pruning are altered in GABA-receptive microglia lacking GABA-B receptors and that mice lacking this receptor develop behavioral abnormalities [42]. About 1/4 of microglia in the murine somatosensory cortex express GABA-B receptors at PD15, and PV boutons are preferentially contacted by these microglia, unless the receptor is selectively deleted from these cells [43]. In another pioneering study, Gallo et al. demonstrated that microglia trigger the formation and maintenance of axo-axonic synapses between inhibitory chandelier cells and the initial segment of axons projected by pyramidal neurons in the somatosensory cortex [41,44]. The authors showed that the deletion of microglial GABA-B receptors decreased the frequency of contacts between microglia and pyramidal neurons and lowered the density of inhibitory initial segment-associated synapses on these neurons [41,44].

Microglia sense environmental changes that may impair homeostasis, affecting their function and morphology. These cells also respond to different stressors through their interactions with neurons, and they impact processes such as adult hippocampal neurogenesis and elimination of synapses [39,45,46]. The perinatal period, which is marked by significant microglial proliferation, maturation and activity, is vulnerable to events that can result in long-term perturbations in the microglial cell number or function [47,48]. Although several reports have demonstrated the role of microglia in different stress paradigms [49,50,51], our knowledge of the effects of early-life experiences on microglial phenotype and function is limited. Intriguingly, most investigations of ELS models have focused on the hippocampus, at least in part because microglia–neuron communication is fundamental for synaptic transmission and plasticity in this brain region [52]. Using brief daily separation, a mouse model of ELS, Delpech et al. reported a perturbed maturation of microglia in the developing hippocampus [53], which led to abnormal maturation of several neuronal and non-neuronal cellular processes, resulting in behavioral abnormalities from the juvenile period throughout adulthood. Interestingly, they showed that microglia isolated from the hippocampus of 28-day-old ELS mice had an increase in phagocytic activity and a decrease in expression of genes that normally increase across development. Furthermore, promoter analysis indicated changes in key microglial transcriptional activity, including PU.1, Creb1, Sp1, and RelA [54]. More recently, using limited bedding, the same group showed synaptic pruning impairment in the developing hippocampus [55]. Of note, limited bedding caused more severe impairment in microglial synaptic engulfment compared to unpredictable postnatal stress [55]. The effects of ELS induced by limited bedding and nesting during the first week of life (PD 2 to PD 9) has also been investigated in the hippocampus at the level of microglial morphology, gene expression, and synaptosome phagocytic capacity in postnatal (PD 9) and adult (PD 200) mice [56]. The hippocampus of ELS-exposed adult mice displayed morphological and transcriptomic changes related to the tumor necrosis factor response. Functionally, synaptosomes from ELS-exposed mice were less phagocytosed than age-matched microglia. At PD 200, but not PD 9, ELS microglia showed reduced synaptosome phagocytic capacity when compared to control microglia [56]. The hypothalamus has also been investigated in the context of ELS-associated microglial changes. Bolton and colleagues showed that ELS induces functional excitatory synapses onto stress-sensitive hypothalamic corticotropin-releasing hormone (CRH)-expressing neurons due to microglial pruning dysfunction [57]. These authors also reported abnormalities in the microglial morphology and function in the hypothalamus, including changes in microglial processes and deficits in phagocytosis. Interestingly, selective chronic chemogenetic activation of microglia in ELS mice with excitatory designer receptor exclusively activated by designer drugs (DREADDs) diminished the excitatory synapse density to control levels and corrected adult acute and chronic stress responses [57]. As mentioned, it is well established that microglia regulate neuroimmune responses and play a crucial role in maintaining homeostatic brain functions [58]. These resident macrophages are functionally one of the most diverse cell types in the CNS, as they dynamically react to their changing environment [58]. Methods such as single-cell RNA sequencing, translating ribosome affinity purification and fluorescence-activated cell sorting (FACS) have helped us gain more information about the heterogeneous nature of microglia and their gene expression profile, which is driven by the brain region, sex, age and type of pathology [35].

## 4. Mechanisms Underlying ECM–Microglia Interactions in Early-Life Stress

The information on the possible interplay between microglia and ECM in the context of child adversity and psychopathologies is still limited, especially in the human brain. Microglia can affect the function and integrity of PNNs through direct and indirect pathways. The interaction between neuronal ligands (from PV^+^ and PV^−^ neurons) and microglial receptors initiates different intracellular cascades that result in the phagocytosis of ECM-PNN components (direct pathway) [59,60,61,62,63]. Microglia also synthesize and release enzymes such as MMP2, MMP9 and Cathepsin-S that regulate the integrity of PNNs through the degradation of these glycoprotein structures (indirect pathway) [64]. In the next few sections, we try to provide a clearer picture of the possible direct and indirect regulation of ECM-PNNs by microglia and hypothesize about the pathways underlying microglia and PNN interactions following ELS.

### 4.1. Direct Modulation of the ECM by Microglia and Possible Regulation by ELS

Recent breakthrough studies have suggested that microglia can regulate PNNs and synaptic integrity in health and neurodegenerative conditions such as AD [59]. Crapser et al. showed that microglial depletion prevents ECM changes and striatal volume reduction in a model of Huntington’s disease [60]. The same group reported that microglia are responsible for the loss of PNNs in the AD brain [61]. These findings provided novel information on the direct regulation of PNNs by microglia in neurodegenerative conditions. Using immunofluorescence, these investigators revealed the close spatial association between morphologically altered PNNs and microglia, which contained inclusions of PNN material, in agreement with phagocytic uptake [61]. Moreover, using microglial depletion, significant changes in the PNN counts were observed. Although these studies clearly provide evidence of the engulfment of PNN components by microglia, little is known about the mechanisms underlying this process. Two more recent investigations have explored microglia–ECM interactions at the molecular level and have involved the IL33 receptor and CX3CR1 signaling [62,63]. Nguyen et al. demonstrated molecular interactions between neurons and microglia that mediate experience-dependent synapse remodeling in the hippocampus [63]. Their findings indicate that the interaction between interleukin-33 (IL-33), expressed by adult hippocampal neurons, and IL-33 receptors, which are exclusively expressed by microglia, is fundamental for ECM remolding. Indeed, neuronal IL-33 triggers microglial engulfment of the ECM, and its loss leads to impaired ECM engulfment and deposition of ECM proteins in contact with synapses [63]. The same study identified aggrecan immunostaining within microglia and found that it localized to CD68^+^ lysosomes. In microglial IL-33 KO mice, decreased numbers of CD68^+^ lysosomes were found compared to controls, indicating that IL-33 promotes phagocytosis [63]. Interestingly, the authors also showed that IL-33 controls the expression of several microglial genes that are involved in ECM regulation, including many ECM proteases (Adamts4, MMP14, MMP25, and Ctsc). For instance, Adamts4 and MMP14 are membrane-bound metalloproteases that mainly cleave CSPGs [63]. Previously, our group demonstrated that CA is associated with increased recruitment of PNNs in the vmPFC [3]. Our more recent work indicates that the IL33R protein level is decreased significantly in this brain region in cases vs. controls. This is the first report in humans implicating microglia in modulating PNN integrity following CA [64].

CX3CR1 has also been associated with microglia–PNN interactions. Tansley et al. investigated the role of microglia in peripheral nerve injury-mediated PNN degradation. They found that mice lacking microglial CX3CR1 have significantly less PNN degradation around lamina I projection neurons and diminished WFA accumulation [62]. As mentioned above, CA is linked to increased recruitment of PNNs in the vmPFC [3]. Using ELISA, we measured CX3CR1 in the grey matter vmPFC of depressed suicides with a history of CA and found, intriguingly, a robust reduction in the protein level of this receptor in the CA group compared to the matched controls [64]. Although we previously observed that the expression of canonical components of PNNs is enriched in oligodendrocyte progenitor cells (OPCs), and that they are upregulated in child abuse victims, our more recent findings imply that microglia might also play important roles in regulating PNN via CX3CR1-mediated phagocytosis. It can be hypothesized that the expression or degradation of neuronal CX3CL1 (a selective ligand for microglial CX3CR1) is also altered by CA. Interestingly, we found a lower expression level of Cathepsin S in the vmPFC of CA victims [64]. Cathepsin S, a member of the cysteine cathepsin protease family, is a lysosomal protease engaged in PNN degradation [65,66]. The soluble chemokine CX3CL1 (fractalkine), on the other hand, is cleaved from membrane-bound CX3CL1 by cathepsin S. Therefore, a CA-associated lower expression of Cathepsin in the vmPFC might have led to lower levels of soluble CX3CR1. This would mean that in the context of CA, the level of the active form of neuronal CX3CL1, microglial CX3CR1, or both, is decreased.

An important open question concerns the capability of microglia to synthetize PNN components. To answer this question with a comprehensive view, we examined the snRNA-sequencing findings previously generated by our group in the dlPFC of neurotypical subjects (Figure 1). We performed cell-type enrichment analysis [67] for several PNN component genes. Interestingly, this approach revealed that OPCs and astrocytes, but not microglia, are enriched for some of the canonical PNN components. For example, the OPC cluster highly expresses *VCAN*, *PTPRZ1* and *TNR*. Astrocytes also express high levels of *NCAN*, *BCAN* and *PTPRZ1*.

### 4.2. Indirect Modulation of the ECM by Microglia and Possible Regulation by ELS

Microglia secrete several proteases that are capable of degrading the ECM and, more specifically, PNN components. Matrix metalloproteinases (MMPs) and their endogenous inhibitors, tissue inhibitors of metalloproteinases (TIMPs), are among the most important enzymes contributing to microglia-mediated PNN regulation [59]. Different types of MMPs, especially MMP2 and MMP9, are expressed by microglia and their upregulation has been reported in neurological conditions such as brain ischemia and multiple sclerosis, in which PNN degradation occurs [70,71,72]. Studies reveal that MMP9 released by glia and neurons is directly responsible for PNN degradation, as evidenced by the increased densities of PNNs following genetic deletion of the MMP9 gene in Fmr1 knockout mice, a model of fragile X syndrome [73]. Intriguingly, the pharmacological effects of the antidepressant venlafaxine in mouse stress paradigms have also been correlated to PNN degradation by MMP9 [74].

We recently investigated the potential role of microglia in the CA-associated increase of PNNs observed in the mature vmPFC with comprehensive unbiased techniques such as the human matrix metalloproteinase array with vmPFC grey matter total lysates and with isolated CD11b-positive microglia [64]. We observed that the levels of MMP1, MMP3, MMP8, MMP9, TIMP1 and TIMP2 were decreased in the grey matter in samples from individuals with a history of CA, suggesting less MMP-mediated PNN degradation or cleavage. Furthermore, our results revealed a significant CA-associated downregulation of MMP9 and tissue inhibitors of metalloproteinases TIMP2 in microglia isolated from the vmPFC [64]. MMP9 has been implicated in major depression on the basis of evidence that this enzyme interferes with several processes, such as inflammatory responses, blood-brain barrier permeability, regulation of PNNs, demyelination and synaptic long-term potentiation [75]. A novel ribosome-based regulatory mechanism/checkpoint in mice has been described recently, which controls the innate immune gene translation and microglial activation in non-sterile inflammation and following brain ischemia, orchestrated by RNA binding proteins [76,77]. Interestingly, all the MMPs and their endogenous inhibitors mentioned above are expressed in microglial-translating mRNA [76,77]. This confirms that microglia are an important source of various MMPs in the brain. Our post mortem findings revealed that microglia might be involved in PNN regulation in CA through indirect MMP-mediated mechanisms [64]. In the previous section, we discussed the importance of IL33-mediated microglial modulation of PNNs. Interestingly, IL33 can indirectly affect ECM integrity through the induction of MMP14 and MMP25 [63]. It will be important, in future studies, to measure these two MMPs, given that we found lower IL33R levels in the vmPFC of CA victims.

Cathepsin S secreted from microglia is involved in the diurnal variation of dendritic spine density borne by cortical neurons though the proteolytic modification of perisynaptic ECM molecules [78,79]. Intriguingly, Hayashi et al. reported that cortical microglia contain an intrinsic molecular clock and exhibit a circadian expression of Cathepsin S. The deletion of Cathepsin S causes mice to display increased locomotor activity and eliminates diurnal variations in the activity and spine density of cortical neurons, which are significantly higher during the dark phase than the light phase [80]. PNNs have been shown to respond dynamically during learning, potentially regulating the formation of new synapses. It has been hypothesized that PNNs vary during sleep, a period of active synaptic modification. In addition, PNN components can be cleaved by Cathepsin S, coinciding with dendritic spine density rhythms. In this context, Cathepsin S may contribute to PNN remodeling during sleep, mediating synaptic reorganization [66]. Pantazopoulos and colleagues reported diurnal and circadian rhythms of PNNs labeled with the lectin WFA in multiple mouse brain regions involved in emotional memory processing. Diurnal rhythms of Cathepsin S expression in microglia were observed in the same brain regions and were the opposite to the diurnal pattern of PNN expression [66]. We recently found that the Cathepsin S levels are significantly lower in the vmPFC of depressed suicides with a history of CA and who died at night [64]. It should be mentioned that we have not thoroughly studied the diurnal fluctuations of the PNN densities in samples from this group. Future investigations will explore these variations in parallel to those displayed by microglial Cathepsin S expression.

## 5. Interactions between Microglia and Other Glial Cell Types Following ELS Modulate PNN Integrity

Microglia interact with neurons but also with other glial cells, such as astrocytes, OPCs and mature oligodendrocytes. Such interactions are also likely to affect the ECM in the short and long term following ELS. For example, microglia are fundamental for proper oligodendrocyte development, proliferation, and maturation due to secreting proteins such as VEGF and insulin-like growth factor (IGF)-1 [81], a phenomenon that could be implicated in the OPC-mediated PNN abnormalities recently observed in depressed suicides with a history of CA [3]. In that study, canonical components of PNNs were found to be highly expressed in OPCs and to be positively correlated with PNN densities in victims of CA, suggesting that early-life adversity can substantially affect the expression of OPC genes that are key to PNN development, maturation and maintenance [3].

Microglia also interact substantially with astrocytes. For example, developing astrocytes strongly express IL33, a cytokine crucial for microglial synapse engulfment and neural circuit development through its binding to microglial IL33 receptors [82], as discussed above. Post mortem studies and animal models of chronic stress have consistently associated it with changes in the number and function of astrocytes [83]. This astrocytic vulnerability more than likely disrupts IL33 signaling, which would also impact PNNs. Figure 2 summarizes the different interactions between microglia and other glial cells that are relevant to the modulation of ECM-PNNs in inhibitory neurons. These interactions can be significantly altered by CA.

## 6. Conclusions

In conclusion, microglia are pivotal homeostatic regulators of ECM-PNNs under physiological conditions, and they can display functional adaptations in response to stressors. Histological and transcriptional studies strongly suggest that ELS has profound effects on microglial morphology (e.g., retraction of processes) and function (phagocytotic capacity and secretory profile). Meanwhile, ELS has been associated with both increasing and decreasing the density of PNNs depending on the type of stressor, sex of the animal or brain region investigated (Table 1). The complexities of microglia–PNNs interactions are only beginning to be uncovered in the healthy brain and have yet to be comprehensively studied in the context of ELS in different brain regions and in a sex-specific manner. Furthermore, although animal models of ELS, such as limited bedding and nesting, are useful to understand the mechanisms underlying microglia–PNNs interaction, they cannot fully reflect the complexity of the human brain and experience. It therefore becomes essential to study the cellular and molecular consequences of childhood adversity in well-characterized post mortem human brain samples in different brain regions and in a sex-specific manner to better understand the long-lasting effects of ELS.

ELS seems to influence microglia–ECM interactions through direct and indirect pathways. As discussed in this short review, PNN protein components have been observed within microglia in both mice and humans, suggesting that microglia phagocytose these specialized ECM structures directly. Following ELS, microglial phagocytotic activity and expression of enzymes involved in PNN degradation can be strongly altered. There is growing evidence that microglia can trigger the loss of PNNs, thereby possibly contributing to altered cerebral function and neuroplasticity in psychopathologies, and particularly, in depression. More work is needed, however, to characterize the dysregulated signaling pathways underlying ELS-mediated dysfunction in microglia–neuron, microglia–astrocyte, and microglia–oligodendrocyte communication. Knocking out microglia-specific genes involved in direct (IL33R and GABA-B receptor) or indirect (MMP2 and MMP9) pathways would provide valuable information regarding PNN–microglia interplay following ELS. Moreover, using recent high-throughput techniques such as spatial transcriptomics in post mortem brain sections would inform about the gene expression profiles of microglia surrounding PV^+^ and PV^−^ interneurons enwrapped (or not) by PNNs. This could lead to the identification of key molecular pathways engaged by microglia toward the modulation of PNNs in both healthy and psychopathological conditions. Lastly, potential confounding factors should be carefully considered in such studies as sex, for example, can affect both innate immune responses and PNN densities in the brain [35,84,85,86].

## Figures and Tables

**Figure 1 biomolecules-14-01087-f001:**
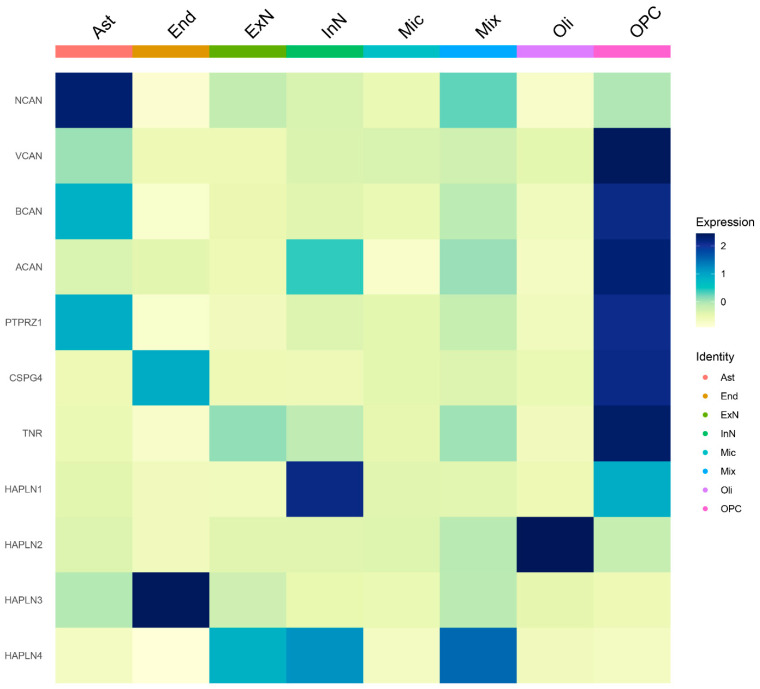
Cell-type enrichment analysis of PNN components in the dorsolateral prefrontal cortex (dlPFC) of healthy subjects. Heatmap showing the expression levels of PNN canonical components derived from post mortem dlPFC samples from 34 neurotypical individuals (16M/18F) (Maitra et al., 2023) [68]. The OPC cluster highly expresses *VCAN*, *PTPRZ1* and *TNR*. Microglia cells barely express—if at all—canonical PNN components such as *VCAN*, *ACAN*, *BCAN* and *TNR*. The processed dataset by Maitra et al. (2023) [68], including the cluster annotations, was downloaded through the UCSC Cell Browser with the following link: https://cells.ucsc.edu/?ds=dlpfc-mdd (accessed on 30 June 2024) Using the R package Seurat [67] (v. 4.1.1, R version 4.1 [69]), the matrix.mtx, features.tsv, barcodes.tsv and meta.tsv files were loaded into a Seurat object. Only neurotypical subjects (*n* = 34) were included in the following analysis. The heatmap was generated using Seurat’s DoHeatmap function with default parameters and the following set of genes inputted as features to include: *NCAN*, *VCAN*, *BCAN*, *ACAN*, *PTPRZ1*, *CSPG4*, *TNR*, *HAPLN1*, *HAPLN2*, *HAPLN3*, *HAPLN4*. All the PNN components were derived from Tanti et al. (2022) [3]. The expression values (normalized expression) of each marker were scaled and are presented as z-scores in the colour bar of the heatmap. Expression is grouped by broad cell types. A darker colour (blue) indicates higher levels of expression. Abbreviations: Ast, Astrocytes; End, Endothelial cells; ExN, Excitatory neurons; InN, Inhibitory neurons; Mic, Microglia; Mix, Mixed expression profile; Oli, oligodendrocytes; OPCs, Oligodendrocyte precursor cells; PNN, Perineuronal net.

**Figure 2 biomolecules-14-01087-f002:**
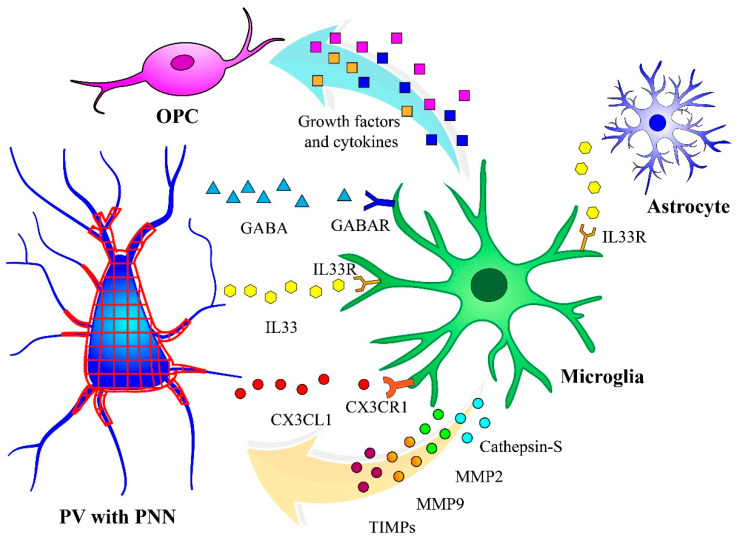
Microglial interactions with other glia cells that are pertinent to ECM-PNN regulation. Inhibitory neurons release GABA, cytokines (e.g., IL33) and chemokines (e.g., CX3CL1). The receptors for these molecules are highly expressed by microglia. The interplay between these neuronal ligands and microglial receptors initiates different signaling cascades that culminate in the engulfment of ECM-PNN components. Furthermore, non-neural cells such as astrocytes can affect the interaction between microglia and PNNs through the release of IL33. Microglia also synthesize and release enzymes such as MMP2, MMP9 and Cathepsin S that indirectly modulate the integrity of PNNs. Finally, microglia secrete several pro-inflammatory and anti-inflammatory cytokines and chemokines. These molecules can robustly regulate OPC proliferation, maturation and survival.

**Table 1 biomolecules-14-01087-t001:** Summary of ELS effects on PNNs and PV interneurons.

Study	Species/Sex	ELS Paradigm/Experience	Timing of Investigation	Brain Region(s)	Effect on PNN/PV
Umemori et al., 2015 [23]	C57BL/6J Mouse Male and Female	Perinatal exposure (GD 7–PD 7) to Fluoxetine or Methylmercury	PD 17 PD 24	Basolateral amygdala Hippocampus (DG and CA1) Basolateral amygdala Hippocampus (DG and CA1)	Flx and MeHg: ↓ PNN density MeHg: ↓ PV density Flx: ↓ PNN density MeHg: ↓ PNN density, ↑ PV density Flx: ↓ PNN density MeHg: ↓ PNN density, ↑ PV density
Santiago et al., 2018 [24]	Long-Evans Rat Male and Female	Scarcity adversity model (PD 8–12)	PD 23	Basolateral amygdala	↓ PNN intensity
Page et al., 2018 [25]	C57BL/6J Mouse Male and Female	Unpredictable mild stress (PD 28–42)	PD 48 PD 80–83	Prelimbic medial PFC Prelimbic medial PFC	↓ PV density in males
Murthy et al., 2019 [26]	C57BL/6J Mouse Male and Female	Maternal separation (PD 2–16) and early weaning (PD 17)	PD 60–70	Ventral hippocampus (DG, CA3 and CA1)	Male DG: ↓ PV density, ↓ PV intensity, ↑ PNN intensity
Gildawie et al., 2020 [27]	Sprague-Dawley Rat Male and Female	Maternal separation (PD 2–20)	PD 20 PD 40 PD 70	Prelimbic PFC Infralimbic PFC Basolateral amygdala Prelimbic PFC Infralimbic PFC Basolateral amygdala Prelimbic PFC Infralimbic PFC Basolateral amygdala	↓ PNN density ↓ PNN density ↓ PNN density ↑ PV density in males ↑ PNN intensity on PV in males ↓ PNN density
Guadagno et al., 2020 [28]	Sprague-Dawley Rat Male and Female	Limited bedding and nesting (PD4–10)	PD28	Basolateral amygdala	↑ PNN density and ↑ PNN intensity in males (right hemisphere only)
* Catale et al., 2022 [2]	DBA Mouse Male and Female	Early-life social stress from CD-1 mouse (PD 14–21)	PD60-70	Anterior cingulate cortex	↑ PNN density
Tanti et al., 2022 [3]	Human Male and Female	Severe child abuse (<15 years old)	34–50 years old	Ventromedial PFC	↑ PNN density
Théberge et al., 2024 [29]	Human Male and Female	Severe child abuse (<15 years old)	35–57 years old	Ventromedial PFC	↑ PNN density on unmyelinated PV, ↑ PV intensity of myelinated PV with PNN
** Belliveau et al., 2024 [30]	Human Male and Female C57BL/6J Mouse Male and Female	Severe child abuse (<15 years old) Limited bedding and nesting (PD 2–9)	28–68 years old PD 70	Ventromedial PFC Hippocampus Entorhinal cortex Medial PFC Ventral hippocampus Entorhinal cortex	No changes in sulfation code or labeling pattern No changes in sulfation code or labeling pattern

All studies labeled PNNs with Wisteria Floribunda Agglutin except for the study marked with * in the first column, which labeled the core protein aggrecan. ** Indicates the only study where WFA and aggrecan were both labeled. Only statistically significant results of ELS compared to controls are highlighted in the last column. ELS: Early-life stress; PNN: Perineuronal net; PV: Parvalbumin-positive neuron; GD: Gestational day; PD: Postnatal day; DG: Dentate gyrus; CA1: cornu ammonis 1; Flx: Fluoxetine; MeHg: Methylmercury; PFC: Prefrontal cortex; DBA: DBA2/J @Ico mice; ↑: increase; ↓: reduce.

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
