# Peer review of "Perineuronal Net Alterations Following Early-Life Stress: Are Microglia Pulling Some Strings?"

_biomolecules, 2024, doi:10.3390/biom14091087_

Round 1

Reviewer 1 Report

Comments and Suggestions for Authors

The brief review by Rahimian et al. discusses the importance of the extracellular matrix (ECM) in the effects of child abuse. The article's strength is the inclusion of post-mortem clinical cases, which are rarely included in the discussion in the child abuse literature. The authors have certainly defined their niche. The review covers relevant clinical and preclinical data regarding the role of microglia, pruning, and PNNs. The inclusion of a summary figure for a review article is also appreciated.  Overall, the manuscript is focused and interesting.

Major comments:

The title is a bit misleading. The manuscript is primarily focused on PNNs and microglia- so why not include these key words in the title?

When the authors refer to the extracellular matrix (ECM), do they mean PNNs (lines 30-31)? This is an important point that needs to be clarified right at the beginning of the manuscript. 

Page 2, lines 68-73: This is a nice quantitative description of the cell types wrapped in PNNs. However, the math is quite shy--I would like to know about the remaining cell types in the vmPFC that are covered by PNNs.  Only 26% are discussed.

Page 2, line 93: is there a study that used a chABC to destroy the PNN in ELS animals?

Page 3, line 106: to what brain region are you referring?

Please check carefully the findings from Gildawie et al. and clarify the comments:

                  Line 111: the prelimbic cortex is  considered part of the prefrontal cortex.

                  Line 113-114: The statement about Gildawie's findings in the infralimbic cortex are incorrect as stated. There was no change in density in the IL.

Line 132: please qualify the study as preliminary given where it is published, and not peer-reviewed (or update the reference).

Line 148: what developmental period are you referring to? Region-dependent?

Minor:

Page 1, line 34: please fix this awkward sentence.

Page 3, line 101: please fix this awkward sentence. "A 2018 study using a develop- 101 mental scarcity-adversity mouse model associated an enhanced threat response from the..."  try "found" instead of "associated"

Section 3: Please fix the font /size of font change.

Line 278, 381: PNN, not PPN

Reviewer 2 Report

Comments and Suggestions for Authors

In this review article, Rahimian and colleagues aimed at reviewing and integrating the existing literature regarding the dynamics of microglial regulation of the ECM and how it can be impacted by early life stress. The collected bibliography encompasses animal models and human post-mortem studies, and suggests possible direct and indirect mechanisms through which microglia may regulate the ECM/PNN and how this can be affected by ELS.

This is an interesting and well-written review that is relevant not only for the field of ELS-Microglia-PNN research, but also for ELS and PNN research only. I appreciated the effort made to integrate data from different studies to provide models of ELS-Microglia-PNN interaction, even though causative data are scarce.

I report some concerns (and suggestions) about the text structure that, if addressed, would make the text more easily accessible. I also have other minor comments on spelling.

Text structure comments:

1.      Paragraph 2: ECM-PNN in development and early life stress:

·        I think it would be really helpful to have a recap table of the effects of ELS on PNN development. It would definitely make the information in this paragraph easier to access. It’s not compulsory, but I believe it would increase the impact of the review, especially because (to my knowledge) there is not another recent review on ELS and PNN. The table could contain info on the species, type of adversity, period of exposure, effect on PNN, marker of PNN, etc.

2.      Paragraph 3: Microglia in development and early life stress

·        Lines 147-151 – why is it relevant to talk about the axon tract-associated microglia (ATM) state? Because it is not clear in this paragraph and it does not come up again in the text.

·        Lines 168-70: I think this sentence could be misplaced, and it’s not very clear here. I would suggest the Authors put it at line 154, after the sentence: “It is noteworthy that most of our knowledge on the importance of microglia during development stems from studies on excitatory neurons [39,40], with only recent work having been conducted on microglial interactions with inhibitory neurons, such as parvalbumin (PV) interneurons [41].”

·        Line 182: “…Delpech et al. reported a perturbed maturation of microglia…” Please indicate briefly how microglia were perturbed in this study (morphological, transcriptional, protein expression, phagocytosis...)

3.      Paragraph 4: Mechanisms underlying ECM-microglia interactions in early life stress

·        Lines 208-215: From “These resident macrophages are…” to “…microglial transcriptome and function under basal and immune-challenged conditions [56].” - I believe this whole paragraph belongs to the previous section on microglia and ELS.

·        At the end of the paragraph, lines 224-225, the Authors state: “Over the next sections, we tried to provide a clearer picture of the possible direct and indirect regulation of ECM-PNNs by microglia.” However, they not only do that, but also provide valuable hypothesis on possible pathways underlying microglial effect on PNN in ELS. I would stress this point at the end of this introductory paragraph, and also in the titles of the subparagraph (e.g., “4.1. Direct modulation of ECM by microglia” could become “4.1. Direct modulation of ECM by microglia and possible regulation by ELS”).

·        Line 278: “An important open question regards the capability of microglia to synthetize PPN components”. Existing mouse databases already suggest that microglial synthesis of PNN components is lower/null compared to other cell types (see Brain RNA-Seq project/Zhang et al. (2014) J Neurosci, Zhang et al. (2016) Neuron, Bennett et al. (2016) PNAS). Although I find the new data provided by the Authors very relevant (since they show/confirm that even human microglia cannot express PNN components), I am concerned about publishing new data in a review.

4.      Paragraph 5:

·        Title: “Interactions between microglia and other glial cell types” I would add “in the modulation of ECM after ELS/that are pertinent to ECM-PNN regulation” or something that refers to the fact that you are focusing on microglia-other glial cell interactions that are relevant for ELS.

5.      Paragraph 6: conclusions

·        Authors talk about how “Following ELS, microglial phagocytotic activity and expression of enzymes involved in PNN degradation can be strongly altered.”, but the conclusion lacks a real conclusive sentence/subparagraph that recaps the different effects of ELS on MG and PNN (increase? Decrease? Change in structure?) and the limitations of the studies. For example, these effects can be influenced by the same variables that influence the research on ELS and PNN, as stated in paragraph 2 (model organism/ELS paradigm/brain region/PNNs marker) as well as the way microglia are studied (morphology/expression/function/state).

·        Lines 405-407: The sentence: “Previous investigations have revealed that activated microglia express both MMP2 and MMP9. In particular, MMP9 is co-localized with NMDA and AMPA receptors and might be involved in synaptic plasticity [67-69].” belongs more to the paragraph 4.2, e.g., at line 317, along with an explanation as to why this may be relevant to PNN.

Minor comments:

Line 38: ELA is used instead of ELS

Line 278: PPN is used instead of PNN
